# MSRNet: Multiclass Skin Lesion Recognition Using Additional Residual Block Based Fine-Tuned Deep Models Information Fusion and Best Feature Selection

**DOI:** 10.3390/diagnostics13193063

**Published:** 2023-09-26

**Authors:** Sobia Bibi, Muhammad Attique Khan, Jamal Hussain Shah, Robertas Damaševičius, Areej Alasiry, Mehrez Marzougui, Majed Alhaisoni, Anum Masood

**Affiliations:** 1Department of CS, COMSATS University Islamabad, Wah Campus, Islamabad 45550, Pakistan; sobia.bibi@ciitwah.edu.pk (S.B.); jhshah@ciitwah.edu.pk (J.H.S.); 2Department of Computer Science and Mathematics, Lebanese American University, Beirut 1102-2801, Lebanon; attique.khan@hitecuni.edu.pk; 3Department of CS, HITEC University, Taxila 47080, Pakistan; 4Center of Excellence Forest 4.0, Faculty of Informatics, Kaunas University of Technology, 51368 Kaunas, Lithuania; robertas.damasevicius@ktu.lt; 5College of Computer Science, King Khalid University, Abha 61413, Saudi Arabia; areej.alasiry@kku.edu.sa (A.A.); mhrez@kku.edu.sa (M.M.); 6Computer Sciences Department, College of Computer and Information Sciences, Princess Nourah Bint Abdulrahman University, Riyadh 11564, Saudi Arabia; mmalhaisoni@pnu.edu.sa; 7Department of Circulation and Medical Imaging, Faculty of Medicine and Health Sciences, Norwegian University of Science and Technology (NTNU), 7034 Trondheim, Norway

**Keywords:** skin cancer, contrast enhancement, deep learning, feature selection, classification, marine predator optimization, fusion

## Abstract

Cancer is one of the leading significant causes of illness and chronic disease worldwide. Skin cancer, particularly melanoma, is becoming a severe health problem due to its rising prevalence. The considerable death rate linked with melanoma requires early detection to receive immediate and successful treatment. Lesion detection and classification are more challenging due to many forms of artifacts such as hairs, noise, and irregularity of lesion shape, color, irrelevant features, and textures. In this work, we proposed a deep-learning architecture for classifying multiclass skin cancer and melanoma detection. The proposed architecture consists of four core steps: image preprocessing, feature extraction and fusion, feature selection, and classification. A novel contrast enhancement technique is proposed based on the image luminance information. After that, two pre-trained deep models, DarkNet-53 and DensNet-201, are modified in terms of a residual block at the end and trained through transfer learning. In the learning process, the Genetic algorithm is applied to select hyperparameters. The resultant features are fused using a two-step approach named serial-harmonic mean. This step increases the accuracy of the correct classification, but some irrelevant information is also observed. Therefore, an algorithm is developed to select the best features called marine predator optimization (MPA) controlled Reyni Entropy. The selected features are finally classified using machine learning classifiers for the final classification. Two datasets, ISIC2018 and ISIC2019, have been selected for the experimental process. On these datasets, the obtained maximum accuracy of 85.4% and 98.80%, respectively. To prove the effectiveness of the proposed methods, a detailed comparison is conducted with several recent techniques and shows the proposed framework outperforms.

## 1. Introduction

The most deadly kind of skin cancer, melanoma, has increased dramatically worldwide. Consequently, early and prompt diagnosis is crucial for reducing the severity of the disease. The analysis of medical images of different organs of the body to detect irregular behavior plays a vital role in the medical field, such as skin cancer [1], brain cancer [2], lung cancer [3], breast cancer [4], and retina [5]. Skin cancer is one of the more prevalent diseases today [6]. It is one of the most common forms of cancer in humans because it is the body’s largest organ [7]. The skin lesion is generally divided into two classes, i.e., melanoma and non-melanoma [8]. The World Health Organization (WHO) reports that there were 104,350 cases of skin cancer overall and 11,650 fatalities in the United States in 2019 [9]. In 2020, 196,060 new cases of skin cancer are anticipated. It is believed that 40,160 and 60,190 of the latter are men and women, respectively [10]. Based on these figures, it is possible to anticipate that in 2020, the situations will more than triple while the death rate will decrease by over 5.3%. In the United States, 106,110 new instances of melanoma are anticipated to be diagnosed in 2021, while 7180 people will pass away from the disease.

Melanocytes are the cells in which melanoma develops when these cells overgrow and form a malignant tumor [11]. The hands, face, neck, lips, and other exposed skin parts are particularly affected by it [12]. Early detection of melanoma increases the likelihood of being successfully treated; otherwise, it will spread to other body areas and cause an agonizing death [13]. After an eye exam, it might be challenging for specialists to diagnose skin cancer in its early stages [14] as modern specialized, computer-aided detection (CAD) technology has been employed to identify all types of tumors since the early 2000s [15].

Melanoma includes complex patterns of multiple components and exhibits asymmetrical pigment distribution on the acral skin. The blue nevus (blue-grey region) aids in detecting malignancy, whereas these pigment networks, dots, or globule distributions help identify melanocytic diseases. Any lesion that does not exhibit the traits above is said to be non-melanocytic.

Dermoscopy, a non-invasive imaging technique, has been created to assist dermatologists in their clinical examination to effectively diagnose melanoma [16]. Due to good visual perception, the dermoscopy device can be useful for discriminating between malignant and benign skin lesions. The capacity of dermatologists to discriminate between melanoma and non-melanoma images has been improved by the development of several traditional approaches, including the ABCD rule [17], 7-point checklist [18], Menzies procedure [19], and CASH [20]. Due to intra-class similarity, an expert person’s accurate diagnosis of skin cancer is challenging. Furthermore, melanoma and non-melanoma skin cancer kinds are very similar in color, size, and other characteristics.

Additionally, eye examination-based melanoma diagnosis is laborious, expensive, and time-consuming [21]. Hence, developing a computerized technique for accurately diagnosing and classifying skin cancer is very important. Several computerized techniques have been introduced in the literature for detecting and classifying skin cancer. A computerized technique is based on a few important steps such as preprocessing the dermoscopic images, lesion detection, feature extraction, and classification. Deep learning (DL) techniques have successfully detected and classified cancer diseases in medical imaging [22,23]. For the skin cancer classification, DL techniques give promising results that reveal its importance in medical imaging [24].

### 1.1. Motivation

The skin is the largest and most important organ in the human body. Skin cancer is currently the most common and deadliest type of cancer. It is a very specific area of research in image processing and computer vision [25]. As was previously mentioned, melanoma is the cancer that causes the greatest destruction and spreads the fastest worldwide. The exceedingly complicated makeup of the lesion makes a clinical diagnosis a poor choice. Despite extensive research and the development of numerous techniques, the issue of accurately detecting and classifying skin lesions remains difficult. The primary objective of this research is to develop a trustworthy computer-based melanoma detection technique that can surpass existing computer-aided detection methods [26].

### 1.2. Problem Statement

Advanced machine learning techniques like deep learning are frequently applied in medical imaging for detection and classification. Experts are actively studying skin cancer, and computer vision experts have developed several strategies. Many obstacles make skin lesion segmentation and classification less accurate. This scientific project faces several obstacles, including Low-contrast skin lesions, variations in lesion shape, and irregularity, which degrade the performance of accurate feature extraction. Imbalanced skin classes increase the probability rate of a higher number of image classes that impact the prediction performance of other classes. The researcher occasionally combined data from multiple sources to improve forecast accuracy, but this process significantly influenced the system’s calculation time. Redundant and irrelevant features increase the mistake rate and testing time during training and testing. Furthermore, melanoma, akiec, and nevi were all mistaken for one another during the prediction process. For an accurate multiclass classification problem, adding hidden layers to a neural network or other classifier is always difficult.

### 1.3. Major Contributions

The major contributions of this work are as follows:A contrast enhancement technique is proposed based on the luminance channel and Retinex Model. The proposed technique enhanced the quality of contrast between infected and healthy regions.Fine-tuned two pretrained models and added residual blocks at the end for better learning on the selected datasets.Proposed a serial-Harmonic mean fusion techniqueWe developed an optimization technique named Marine Predator controlled Reyni Entropy for best feature selection.

## 2. Related Work

Nowadays, traditional clinical methods for melanoma diagnosis are ineffective. There is room for a CAD system to classify skin cancer accurately [27]. Colored skin lesions are examined and researched via a method called dermoscopy. It showed a new aspect of skin lesions, enabling diagnostic tools to accurately differentiate between melanoma and non-melanoma lesions. A computer uses dermoscopy to accurately diagnose and categorize skin abnormalities [28,29,30]. The next four important processes are preprocessing, lesion segmentation, feature extraction, and lesion classification. There are a lot of unanswered issues when it comes to accurately detecting and classifying skin lesions.

Deep learning models can be used to optimize the efficiency and quality of skin cancer classification [21]. According to previous literature, the most common approach in dermoscopic Image Analysis (DIA) since 2015 is a convolutional neural network used as a classifier. The latest Advanced computer vision and digital image processing research have revealed the significance of deep learning techniques to attain excellent accuracy in image segmentation, detection, and classification in complex problems [31]. To identify malignant lesions, Codella et al. [32] studied and presented mostly used deep neural networks, such as deep residual networks and deep convolutional neural network models. Simon et al. [33] presented a Deep Learning structure for skin lesion segmentation and classification. The main strength of this work was categorizing the tissues into 12 dermatologist classes. After that, they trained a deep CNN using these characteristics for final classification. They tested the introduced framework on dermoscopy images and compared it with clinical accuracy. During the comparison phase, the clinical method achieved an accuracy of 93.6, whereas the computerized method attained 97.9%. This shows that the computerized methods would perform better than the clinical techniques. Amin et al. [34] introduced an integrated design for deep feature fusion through preprocessing, segmentation, and feature extraction; firstly, they resized the images and converted RGB into luminance channel, then they used the Otsu algorithm and Biorthogonal 2-D wavelet transform to segment the infected part of skin after that pre-trained Alex net and VGG16 use to extract the deep features after that optimal feature is selected by using PCA for classification. Al.masni et al. [35] suggested a frequently used deep learning framework, merging both segmentation and skin lesion classification phases. They utilized a resolution convolutional network (FRCN) to perform the segmentation process over dermoscopic images. After that, different classifiers Inception-v3, ResNet-50, and Inception-ResNet-v2, are used over segmented images. The proposed structure of the deep learning model is experienced by three different dataset ISIC2016, ISIC2017 and ISIC2018 which hold two, three or seven classes of skin lesion with highly balanced, segmentation, and augmentation. The classifiers of Inception-v with 377.04%, ResNet-50 with 79.95%, Inception-ResNet-v2 with 81.79%, and DenseNet-201 with 81.27% showed their predicted accuracies for the dataset of ISIC2016. ResNet-50 outperformed ISIC 2017 in three classes (81.2%, 81.5%, 81.3%, and 73.4%), and ISIC2018 in seven classes (88.05%, 89.28%, 87.74%, and 88.70%), indicating its better performance.

Pacheco et al. [36] used the Thirteen best deep learning networks and observed that the SENet convolutional neural network and Adam optimization are the perfect architecture. The proposed model obtained 91% performance on the ISIC2019 Dataset. The research presented by Farooq et al. [37] enhances the classification performance of 86% of two excellent neural networks, Mobile Net and Inception Net, by utilizing the Kaggle updated dataset of skin cancer. A pioneering-based CNN-based research was conducted by Esteva et al. Lui et al. [38] Proposed a method of categorization of skin lesions; they used a traditional deep learning mode that included Dense Net and Resnet, as well as the MFL module, and achieved an accuracy of 87% on the ISIC 2017 dataset. Pedro et al. [39] introduced a classification model based on Linear SVM and Feedforward Neural Network (FNN), achieving a 90% accuracy on the dermo fit dataset. Milton et al. [40] proposed a comprehensive study of numerous deep learning methods for skin cancer. This study was conducted on many neural networks like Inception Resnet-V2, PNASNet-5, SENet-154, and Inception-V4 on publicly available ISIC-2018 Dataset. The best performance of 76% results was obtained on the PNASNet-5 model. Khatib et al. [41] Resnet-101 Architecture was presented for the classification of skin lesions. On a well-known PH2 database, the suggested model used fine-tuned CNN models to identify the multiple types of skin lesions via transfer learning and achieved an accuracy of 90%. Almaraz et al. [42] used the ABCD rule based on color, shapes, and texture as handcrafted features and Mobile NetV2 neural network architecture by using information measures for the classification of melanoma. The presented technique achieved excellent accuracy of 92.4% on the HAM10000 dataset. Table 1 presented the summary of the few existing techniques.

## 3. Proposed Work

In this section, the proposed method for melanoma classification is presented. The proposed method comprises preprocessing, feature extraction and fusion, feature selection, and classification steps. Figure 1 shows the proposed melanoma classification using deep learning. This figure shows that the deep features are extracted from two pre-trained CNN models, DarkNet-53 and DenseNet-201. The extracted deep features are fused using a novel technique that is later optimized using a feature selection algorithm. The selected features are finally employed for the classification. The description of each step is given in the below sub-sections.

### 3.1. Proposed Contrast Enhancement

#### 3.1.1. Datasets Description

In this work, two datasets have been utilized for the experimental process, such as ISIC2018 [40] and ISIC2019 [44]. Both datasets have been publically available for research purposes (https://challenge.isic-archive.com/data/#2019, accessed on 11 August 2023). The ISIC2018 dataset consists of 10,015 dermoscopic images for training and 1512 testing images. The training images include 1113 of Melanoma (MEL), 6705 of Melanocytic nevus (NV), 514 samples of Basal cell carcinoma (BCC), 327 images of Actinic keratosis (AK), 1099 images of Benign keratosis (BKL), 115 images of Dermatofibroma (DF), and 142 images of Vascular (VASC), respectively.

The ISIC2019 [44] dataset comprises 25,331 training images and 8238 test images. Overall, the total number of images is 33,569. This dataset consists of eight classes: MEL, NV, BCC, AK, BKL, DF, VASC, and SCC (squamous cell carcinoma). All images of both datasets have been in RGB format with different resolutions. We resized all the images into 512 × 512 × 3, which was later resized according to the selected CNN models. A few sample images are shown in Figure 2.

#### 3.1.2. Contrast Enhancement

The lesion diagnosis system’s most crucial phase is contrast enhancement. The issue of low contrast is addressed in the literature using a diversity of enhancing approaches. This article uses a novel technique that uses texture and color information for improvement. Because it is observed that, in contrast to patches of healthy skin, skin lesions are more likely to have texture and color information. The textural information is calculated using normalized luminance channels as follows:(1)$$\varphi_{L}\left(u,v\right)=\left(\lambda \times F\left(Y\right)\right)-16,$$
(2)$$F\left(Y\right)=\left\{\begin{array}{cc} \sqrt[{3}]{Y} & \mathrm{for}\;Y>0.01\\ \left(7.787*Y\right)+\frac{16}{\lambda } & \;\;\;\mathrm{elsewhere} \end{array}\right.$$
where λ = 116, $\left(Y\right)=\frac{\tilde{Y}}{100}$, $\tilde{Y}=\omega_{i}\times G$, $i\in \left\{0.212,0.715,0.072\right\}$. The G denotes the green channel, which is extracted from the original RGB image as $G=\frac{G}{\sum_{j=1}^{3}\phi_{j}}$. The whole expression is simplified as follows:(3)$$L\left(u,v\right)=\varphi_{L}(\sum_{j=1}^{3}\frac{I(u,v)}{3})$$
where $I\left(u,v\right)$ the original RGB is an image and $\varphi_{L}$ is luminance function. Then the Gaussian function is performed on the luminance image to examine the textural information in the lesion area. The Gaussian function is defined as follows:(4)$$\rho \left(u,v,\sigma \right)=\frac{L\left(u,v\right)}{\varphi \left(u,v,\sigma \right)}-L(u,v)$$
where $\varphi \left(u,v,\sigma \right)=L\left(u,v\right)\times G(\sigma )$. It means that $L\left(u,v\right)$ is smoothed by a Gaussian filter with parameter σ (standard deviation). The σ is calculated as follows:(5)$$\sigma =\sqrt{\frac{\sum {\left(uv\right)}^{2}}{N}-{\left(\frac{\sum uv}{N}\right)}^{2}}$$

The above expression $\rho \left(u,v,\sigma \right)$ is simplified as:(6)$$\rho \left(u,v,\sigma \right)=\frac{L\left(u,v\right)-L\left(u,v\right)\times \varphi \left(u,v,\sigma \right)}{\varphi \left(u,v,\sigma \right)}$$
(7)$$=\frac{L\left(u,v\right)\left[1-\varphi \left(u,v,\sigma \right)\right]}{\varphi \left(u,v,\sigma \right)},$$
(8)$$=\frac{L\left(u,v\right)\times Z}{\varphi \left(u,v,\sigma \right)}$$
where $Z=1-\varphi \left(u,v,\sigma \right)$. Generally, the low-intensity pixel in dermoscopic images occurs in the lesion area. Hence, we perform an activation function to differentiate the lesion and skin pixels in the image. The activation function is defined as:(9)$$F(A)=\{\begin{array}{lll} \tilde{\varphi_{L}}\left(u,v\right) & if\rho \left(u,v,\sigma \right)>\varphi (u,v,\sigma ) & \mathrm{Lesion}\;\mathrm{Area}\\ \tilde{\varphi_{H}}\left(u,v\right) & \mathrm{otherwise} & \;\;\mathrm{Healthy}\;\mathrm{Skin}\;\mathrm{Area} \end{array}$$
where, $\tilde{\varphi_{L}}\left(u,v\right)$, $\tilde{\varphi_{H}}\left(u,v\right)$ represents the lesion and healthy skin area, respectively. Finally, to adjust the color intensities of resultant pixels, we utilized the Retinex Model [45]. This model is utilized for color adjustment, which is defined as follows:(10)$$\varphi_{Retinex}\left(u,v\right)=\frac{\tilde{\varphi_{L}^{i}}\left(u,v\right)}{\tilde{\varphi_{L}^{i}}\left(u,v\right)⊗G(\sigma )}$$
where i∈L,A,B; ⊗ denotes the convolution operation and G(σ) is the Gaussian filter with standard deviation. Some sample results of the preprocessing step are shown in Figure 3. This figure clearly shows that the problem of poor contrast is resolved by implementing the proposed technique. These enhanced images are further utilized in the model’s learning phase.

#### 3.1.3. Transfer Learning

Transfer learning (TL) is used to improve the efficiency of the process and reduce the number of resources essential. When elements of a pre-trained machine learning model are reused in a new machine learning model, this is known as transfer learning. In transfer learning, define feature vector and probability distribution as $A=\left\{f_{v},{P(f}_{v})\right\}$ and $f_{v}=\left\{v_{1},v_{2},\ldots ..,v_{n}\right\}$. In which ground truth $G=g_{1},g_{2},\ldots ..,g_{n}$ and objective function ${O=}^{}\{G,l\left(x\right)$, whereas l(x) is an unknown label class. P(g|x) is a probabilistic representation of the function. Transfer learning and the learning rate are denoted as $T_{o}$ and $L_{o}$. $T_{f}$ will be used to show the targeted function and targeted output is $T_{f}$. The main goal of transfer learning is to improve the learning rate for predicting the targeted item using the recognition function ($l\left(x\right)$) depending upon that training for learning from $T_{o}$ and $T_{f}$ where $T_{o}\neq$ $T_{f}$ and $L_{o}\neq T_{f}$. Pattern recognition is improved via inductive transfer learning. You’ll need an annotated database for fast training and testing when using inductive transfer learning. A general model of TL is shown in Figure 4.

### 3.2. Deep Models Fine-Tuning and Feature Extraction

In this work, two pretrained deep learning models, such as DarkNet-53 and DensNet-201 are fine-tuned and trained through TL for deep feature extraction.

**Fine-Tuned DarkNet-53 Model:** A convolutional neural network with 53 layers is known as DarkNet-53 [46]. The ImageNet database contains a pre-trained version of the network trained on more than a million images. This network mainly comprises 53 convolutional layers, 1 × 1 and 3 × 3, located at the front of the residual layer. A batch normalization (BN) layer and a LeakyReLU layer follow each convolutional layer. Several residual blocks of this network are repeated, such as 1, 2, 4, and 8. We deleted the last three layers of the model for the fine-tuning model and added three new layers. In addition, we added a new residual block having three convolutional layers of filter size 3 × 3 and stride 1. After that, the training of this model is performed using TL. After the training, features are extracted from the deeper layer called the global average pool layer of dimensional Nx1024.

**Fine-Tuned DenseNet-201 Model:** DenseNet-201 [47] is the name of a convolutional neural network with 201 layers. A pretrained version of the model that has been tested on more than a million images is present in the ImageNet database. The DenseNet-201 uses the condensed network to produce models that are easy to train and incredibly computationally effective since feature recycling by several layers improves variety in the input to the subsequent layer and performs better.

Figure 5 shows the original architecture of DensNet-201. In the fine-tuning process, we replaced the last three layers at the initial stage with three new layers. After that, a residual block of six layers was added, including three convolutional filter sizes 3 × 3 and stride 1. This block is added after the T3. This fine-tuned model is trained using TL, whereas the global average pooling layer is selected for the deep feature extraction. On this layer, 1920 features are extracted for each image.

### 3.3. Feature Fusion

We are taking two feature vectors $F_{v1}{Dar}_{}$, ${F_{v2}}^{Squ}$ and fusion vector represented as ${Fus}_{v}$. The dimension of these vectors is R×N. Where *N* is represented the length of extracted features and R denotes the number of training images. The initial vector length of each feature vector is R×1024 and R×1920, appropriately. The following formula is used to compute the correlation coefficient between both feature vectors Dar and ${Squ}_{}$ of each row.
(11)$${f\left(Dar\;,Squ\right)=}^{}\frac{COV\;\left(Dar\;,Squ\right)}{\sqrt{Var\left(Dar\right)}\sqrt{Var\left(Squ\right)}}$$

The range of these values lies between (−1,1), where −1 for weak correlation and +1 for strong correlation. The equation of the maximum correlation vector is as follows:(12)$$CV\left(Dar\;,Squ\right)=\varphi \;f\;((m_{1}(Dar),m_{2}(Squ))$$

In this case, φ denotes the Supremum of the overall Borel functions ;Squ: ω→ω which is located between (0, 1). The $CV\left(Dar,Squ\right)$ is the maximum correlation. After that, a harmonic mean-based threshold function is designed for the final fusion as follows:(13)$$H=\frac{n}{\frac{1}{f_{1}}+\frac{1}{f_{2}}+\ldots +\frac{1}{f_{k}}}$$
where H denotes the harmonic mean, f denotes the features of $CV\left(Dar\;,Squ\right)$ and k denotes the feature of a single row. A harmonic mean is used to give a higher weightage of the small value features. The main reason is the reduction of several small value features important for classification. Finally, a threshold function is employed and a fused vector is obtained.
(14)$$Th=\left\{\begin{array}{c} Fusion\;\left(k\right)\;\;\;\;for\;\;{CV}_{k}\geq H\\ Extra\;features\;\left(m\right)\;\;for\;\;{CV}_{m}<H \end{array}\right.$$

The $Fusion\;\left(k\right)$ feature vector is considered for further processing. In this work, a fused vector is obtained of dimension N×2012, where N is the number of training images.

### 3.4. Feature Selection

Feature selection is a hot research area in computer vision for the curse of dimensionality. Many techniques have been introduced in the literature for feature selection for improved accuracy and less computational time. In this work, a metaheuristic algorithm is implemented named the Marine Predator Algorithm (MPA) [48] and modified further with an entropy technique called Reyni Entropy.

The MPA was proposed to mimic the behavior of marine predators in search of Prey, in which the predators use L’evy and Brownian movements as their optimal foraging mechanisms. The velocity ratio *v* of the Prey to the predator is used to make a tradeoff between L’evy and Brownian strategies. When *v* is small or equal to 0.1, the best strategy for the predator is to move in the L’evy steps (exploration phase) regardless of whether the Prey is moving in Brownian or L’evy. However, if *v* is equal to 1, then the best approach for the predator is to move in Brownian steps if the Prey is moving in L’evy steps. Finally, when >10*v*, the predator should not move at all, regardless of whether the Prey is moving in Brownian or L’evy because it will come in itself (exploitation phase). The mathematical model of the MPA is as follows:

*Initialization*: In the first step, the initial solution is uniformly distributed over the search space area using the following formula, where A∈Fusion(k).
(15)$$\vec{x}=A_{\min}+\vec{i}⊗(A_{\max}-A_{\min})$$
where *i*→ is a vector generated randomly within ⊗ represents the entry-wise multiplication, and *A*→*min*, and *A*→*max* are the vectors containing the dimensions’ lower and upper bounds. 

*Elite and Prey matrix construction:* Based on the survival of the fitness theory, the top predator is the one that is best in foraging. Thus, the top predator is used to construct a matrix called Elite.
(16)$$Elite=\left(\begin{array}{ccc} A^{1}{}_{1,1} & A^{1}{}_{1,2}\cdots \cdots & A^{1}{}_{1,d}\\ A^{1}{}_{2,1} & A^{1}{}_{2,1}\cdots \cdots & A^{1}{}_{2,d}\\ A^{1}{}_{N,1} & A^{1}{}_{N,2}\cdots \cdots & A^{1}{}_{N,d} \end{array}\right)$$
where *A*^1^ → represents the top predator vector and is replicated *N* times to build up the elite matrix (*N* is the number of individuals in the population), and *d* is the number of dimensions. This matrix will be updated at the end of each iteration if the top predator is updated. Another matrix, *p*, represents Prey and has the same dimensions as Elite and is used by the predators to update their positions as follows:(17)$$\vec{P}\left(\begin{array}{ccc} A^{1}{}_{1,1} & A^{1}{}_{1,2}\cdots \cdots & A^{1}{}_{1,d}\\ A^{1}{}_{2,1} & A^{1}{}_{2,1}\cdots \cdots & A^{1}{}_{2,d}\\ A^{1}{}_{N,1} & A^{1}{}_{N,2}\cdots \cdots & A^{1}{}_{N,d} \end{array}\right)$$
where $A_{N,d}$ denotes the nth dimensional of d Prey. The optimization process consists of three steps, high-velocity ratio, unit-velocity ratio, and low-velocity ratio. In the high-velocity ratio, the Prey quickly searches the food, and mathematically, it is defined as follows:(18)$$if\;t<\frac{1}{3}t_{max}$$
(19)$$\vec{V_{i}}=\vec{R_{x}}⨂\left(\vec{{Elite}_{i}}-\vec{R_{x}}⨂\vec{P_{i}}\right)$$
(20)$$\vec{P_{i}}=\vec{P_{i}}+F.\vec{N}⨂\vec{V_{i}}$$
where, $\vec{R_{x}}$ denotes the numerical vector, ⨂ denotes the entry-wise multiplication, F denotes the fixed numerical value that is 0.4 in this work, $\vec{N}$ denotes the numerically generated random vector, t is a current iteration, and $t_{max}$ denotes the maximum iterations, respectively.

After that, a unit velocity ratio-based transition stage is considered that is defined as follows:(21)$$if\;\frac{1}{3}t_{max}<t<\frac{2}{3}t_{max}$$

For the first half, the population is calculated as:(22)$$\vec{V_{i}}=\vec{R_{L}}⨂\left(\vec{{Elite}_{i}}-\vec{R_{L}}⨂\vec{P_{i}}\right)$$
(23)$$\vec{P_{i}}=\vec{P_{i}}+F.\vec{N}⨂\vec{V_{i}}$$

For the second half, the population is computed as follows:(24)$$\vec{V_{i}}=\vec{R_{B}}⨂\left(\vec{R_{B}}⨂\vec{{Elite}_{i}}-\vec{P_{i}}\right)$$
(25)$$\vec{P_{i}}=\vec{P_{i}}+F.AP⨂\vec{V_{i}}$$
where AP is an adaptive parameter that is used for the computation of step size as follows:(26)$$AP={\left(1-\frac{t}{t_{max}}\right)}^{\left(2\frac{t}{t_{max}}\right)}$$

In the last step, a low velocity ratio is opted [48]. Then, a FAD is computed for the final prey selection as follows:(27)$$\vec{P_{i}}=\left\{\begin{array}{ll} \vec{P_{i}}+AP\left[x_{min}+\vec{R}⨂\left(x_{max}-x_{min}\right)\right]⨂\vec{B} & \;\;if\;r<0.4\\ \vec{P_{i}}+\left[0.4\left(1-r\right)+r\right]\left(\vec{P_{r1}}-\vec{P_{r1}}\right) & if\;\;r\geq 0.4 \end{array}\right.$$

Here, $\vec{B}$ is a binary vector of value 1 or 0. The Reyni entropy is computed to remove the uncertainty among selected Prey $\vec{P_{i}}$ and then compute the fitness. The Prey, which satisfied the entropy function, is passed for the fitness calculation.
(28)$$Ent\left(\vec{P_{i}}\right)=\frac{1}{1-\alpha }\log\sum_{i=1}^{n}{\vec{P_{i}}}^{\alpha },\;\alpha >1\;and\neq 1$$

Here, Ent denotes the entropy value of each row of selected ith prey. We are using this value in the following for the final selection.
(29)$$Fnc=\left\{\begin{array}{c} \;\vec{Sel}\left(k\right)\;\;for\;\;\vec{P_{i}}\geq Ent\\ ignore,\;\;\;\;Elsewhere \end{array}\right.$$

The selected vector $\vec{Sel}\left(k\right)$ is finally employed for the fitness calculation. This process continues until the number of iterations is completed. In this work, 200 iterations have been selected. After 200 iterations, we got the final feature vector of dimensional N×1768 for ISIC2018 and N×1559 for ISIC2019 dataset, respectively. The selected features are finally classified using machine learning classifiers.

## 4. Experiments and Results

The experimental process of the proposed method is discussed in this section. The proposed method is examined using two different datasets such as ISIC2018 and ISIC2019. These datasets are publicly available for the researchers of medical imaging. Ten classifiers are used to examine the classification accuracy, including Quadratic SVM (QSVM), Wide Neural Network (WNN), Cubic SVM (CSVM), Fine Tree (FT), Gaussian Naive Bayes (GNB), Weighted KNN (WKNN), Cubic KNN (CKNN), Narrow Neural Network (NNN), Bilayered Neural Network (BNN), and Trilayered Neural Network (TNN). The best one is selected based on the highest accuracy value employed for the visual prediction. Each classifier performance is computed based on performance measures such as sensitivity, F1-Score, precision rate, accuracy, FPR, and testing time (sec). Training and testing sets were split before data augmentation into 50:50, meaning 50% of the images in each class were used for training, while the remaining 50% were taken for testing. The validation images are merged into testing images that are utilized for the classification results. The total number of epochs is 100 with a learning rate of 0.0002, momentum of 0.6557, and batch size of 128. All the experiments are evaluated in MATLAB2022b on an Intel Core i7 7th generation CPU possessing 8 GB of RAM and 8 GB graphics card of RTX3060.

### 4.1. ISIC 2018 Dataset Results

The results of this ISIC2018 dataset are presented in four steps. In the first step, fine-tuned DarkNet-53 deep model features are extracted and performed classification. The classification results are given in Table 2. This table shows that the highest noted accuracy is 79.3% of Cubic SVM. The recall rate of this classifier is 49.2%, the sensitivity rate of 72%, the F1-score of 58.6, and the FNR is 27.3%, respectively. Furthermore, the computed time of the Cubic SVM classifier during the testing process is 114.2 s (sec). The rest of the classifiers obtained an accuracy in the range of 55.3–79%.

Table 3 presents the results of DensNet-201 deep features using the ISIC2018 dataset. On this dataset, the obtained highest accuracy of 81.5% by Cubic SVM. The recall rate of this classifier is 53.8%, the sensitivity rate is 74.5%, the F1-score is 62.4%, and FNR is 25.5%. Furthermore, the computational time of the Cubic SVM is 259.6 s (sec). The rest of the classifiers’ accuracy range is between 59 and 81.3%. Compared to the accuracy and other performance measures of both tables (Table 2 and Table 3), it is observed that the accuracy of the DenseNet-201 model is improved than the DarkNet-53 model. However, the DarkNet-53 model is computationally faster than the DenseNet features.

Table 4 shows the proposed fusion results on the ISIC2018 dataset. In this table, quadratic SVM obtained the highest accuracy of 86.2%, while other computed metrics, such as recall rate, precision rate, F1-Score, and last FNR, are 61%, 80%, and 69.2, respectively. The Cubic SVM achieved an accuracy of 86.1%. The computational time of the fusion process is increased, which is a drawback of this step; however, the improvement in accuracy is strength. Compared to the fusion results with Table 2 and Table 3, an almost 5% improvement in the accuracy is observed for Cubic SVM. For the quadratic SVM, the improvement is also above 5%.

The classification results of the proposed feature selection method are given in Table 5. The quadratic SVM obtained the highest accuracy of 85.4%, while other computed measures included a recall rate of 60.8%, precision rate of 78.1%, F1-Score of 68.3%, and FNR of 21.9%, respectively. After the fusion process, the computational time is almost half, as shown in this table. Overall, the selection process maintains consistent accuracy and reduces computational time. The confusion matrix of quadratic SVM is shown in Figure 6 to verify the proposed feature selection performance.

### 4.2. ISIC2019 Dataset Results

The results of this ISIC2019 dataset are discussed in this subsection. Results are computed in several steps, such as fine-tuned DarkNet-19 model features, DenseNet-201 features, fusion, and selection of best features.

In the first step, fine-tuned DarkNet-53 deep model features are extracted and performed classification. The classification results are given in Table 6. This table shows that 98.1% of Cubic SVM is the highest noted accuracy. The recall rate of this classifier is 98.0%, the precision rate is 98.2%, the F1-score is 98.0, and the FNR is 1.8%, respectively. Furthermore, the computed time of the Cubic SVM classifier during the testing process is 267.8 s (sec). The rest of the classifiers obtained accuracy in the range of 56.2–98%.

Table 7 presents the results of DensNet-201 deep features using the ISIC2019 dataset. On this dataset, the obtained highest accuracy of 98.9% by Cubic SVM. The recall rate of this classifier is 98.9%, the sensitivity rate is 98.9%, the F1-score is 98.9%, and FNR is 1.1%, respectively. Furthermore, the computational time of the Cubic SVM is 1177.6.6 s (sec), which is too high. The rest of the classifiers’ accuracy range is between 61.5 and 98.7%. Compared to the accuracy and other performance measures of both tables (Table 6 and Table 7), it is observed that the accuracy of the DenseNet201 model is improved than the DarkNet-53 model. However, the DenseNet-201 model execution time is too high, which is challenging for this method.

After that, the fusion technique is applied, and the results are presented in Table 8. Table 8 shows the proposed fusion results on the ISIC2019 dataset. In this table, quadratic SVM obtained the highest accuracy of 99.1%, while other computed metrics, such as recall rate, precision rate, F1-Score, and FNR, are 99.02%, 99.1%, 99.0, and 0.9, respectively. The computational time of the fusion process is 1329.7 (sec), which is significantly increased than the previous two steps. Compared to the fusion results with Table 2 and Table 3, an almost 1% improvement in the accuracy is observed for Cubic SVM. For the quadratic SVM, the improvement is also above 1%.

The classification results of the proposed feature selection method are given in Table 9. The Cubic SVM obtained the maximum accuracy of 98.9%, while other computed measures include a recall rate of 98.8%, F1 score of 98.8%, and FNR of 1.1%, respectively. Furthermore, the computational time of the Cubic SVM classifier during the testing phase is 655.1 s (sec). Compared with the fusion results, the feature selection technique results are consistent, and time is significantly reduced. Figure 6 shows the confusion matrix of Cubic SVM for the feature selection results. The confusion matrix can be utilized for the verification of proposed results.

### 4.3. Discussion and Analysis

A detailed discussion of the proposed framework has been conducted in this section. In addition, a detailed ablation study is performed to show the importance of each step. Figure 1 shows the proposed model that includes several middle steps. The contrast of both datasets has been enhanced using the proposed technique, discussed in Section 3.1.2. After that, two pre-trained models were trained and obtained the classification results. Later on, fusion is performed and obtains improved accuracy. However, it is also observed that the time was increased during the fusion process. Therefore, a new feature selection technique is developed for better accuracy with less computational time. All the numerical results are discussed in Table 2, Table 3, Table 4, Table 5, Table 6, Table 7, Table 8 and Table 9. In addition, the confusion matrix of both datasets has been illustrated in Figure 6 and Figure 7. These confusion matrixes show how much the correct prediction has been conducted for each class.

Moreover, time is computed for each classifier for all experiments. Based on the noted time in the tables, it is observed that the computational time of the fusion process is significantly increased, which was later reduced by the feature selection technique. Figure 8 shows the accuracy-based comparison of ISIC2019 dataset results after employing the proposed feature selection technique. This figure shows that the accuracy is plotted for four different ratios such as 50:50, 60:40, 70:30, and 80:20, respectively. The average accuracy of all classifiers for the 50:50 approach is 89.81%, whereas for the rest of the combination, the obtained accuracies are 88.68, 89.29, and 89.75%, respectively.

A GradCAM-based visualization is performed for the DenseNet-201 fine-tuned model. This process aims to analyze the newly trained model’s performance. A few sample results are shown in Figure 9. In this figure, it is illustrated that the brown highlighted regions are marked on the cancer region. Figure 10 shows a few sample-labeled images of the entire proposed framework. These images are generated using the proposed method (Cubic SVM classifier). In the end, a brief comparison of the proposed method with several existing techniques has been conducted. Table 10 presents several techniques for comparison with existing methods. In [49], the authors used the ISIC2018 dataset for the experimental process and obtained an accuracy of 83%. The proposed method shows an improved accuracy of 85.4%. Authors in [50,51] used the ISIC2019 dataset and obtained an accuracy of 97.1% and 97.84%, respectively. The proposed method obtained an accuracy of 98.9%, which is improved than the existing techniques on the ISIC2019 dataset.

## 5. Conclusions

Skin lesion classification is vital in computer-aided melanoma detection (CAD) systems, whose accuracy depends on the middle steps, such as contrast enhancement of skin lesions, feature extraction, feature fusion, and selection. This work proposes a non-invasive computerized dermoscopy technique for the improved classification accuracy of multiclass skin lesions. Data augmentation was performed in the initial phase that followed the learning of fine-tuned deep learning models. Features are extracted from the global average pooling layer of both trained models. Later on, the fusion technique is employed, and fused features of both CNN models. Finally, the fused feature vector is optimized using an improved selection algorithm that is classified using machine learning classifiers. Two datasets have been employed for the experimental process, such as ISIC2018 (seven classes) and ISIC2019 (eight classes). On these datasets, the proposed method obtained an improved accuracy of 85.4% and 98.9%, respectively. Overall, we conclude the following:The proposed framework can be useful in the clinics for the second opinion of malignant and benign lesions.The proposed framework can help dermatologists with early classification of lesion type and is also useful for lesion location localization (GradCAM).The contrast enhancement step improves the visibility of cancer and healthy regions, later helpful in better learning of fine-tuned deep models.Adding a new block for each network increased the learning performance and training accuracy.The fusion process improved the accuracy of the proposed method compared to the fine-tuned models.The selection of best features removed the redundant and irrelevant information and reduced the computational time.

This work’s limitation is the increased computational time after employing the fusion step. In the future, an attention mechanism-based network-level fused architecture will be designed and trained on the ISIC2018 and ISIC2019 datasets. In addition, a feature optimization technique will be proposed based on the location adjustment.

## Figures and Tables

**Figure 1 diagnostics-13-03063-f001:**
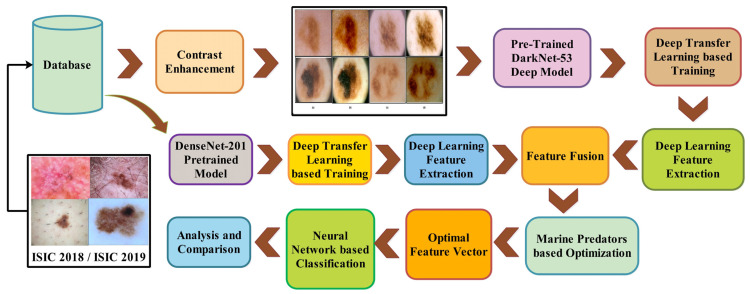
Main flow of proposed automated melanoma recognition using deep learning.

**Figure 2 diagnostics-13-03063-f002:**
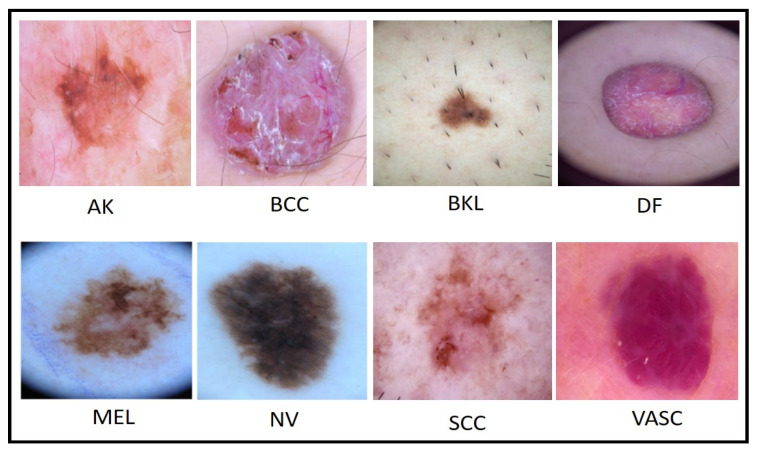
A sample image of the ISIC2019 dermoscopic dataset.

**Figure 3 diagnostics-13-03063-f003:**
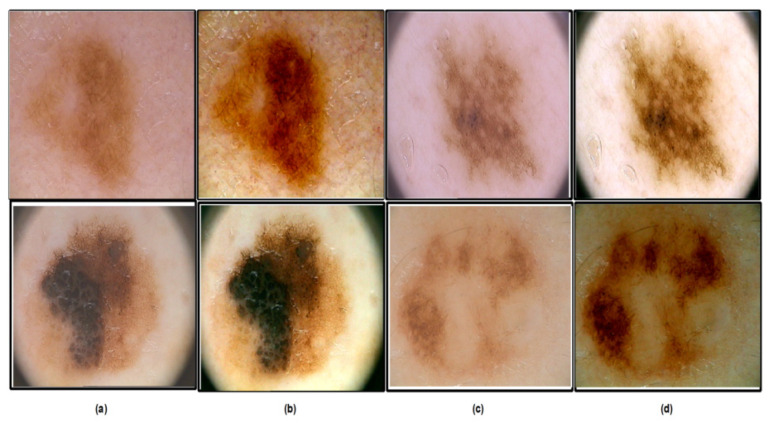
Lesion Contrast Enhancement Results: (**a**,**c**) Original Image; (**b**,**d**) Enhanced Image.

**Figure 4 diagnostics-13-03063-f004:**
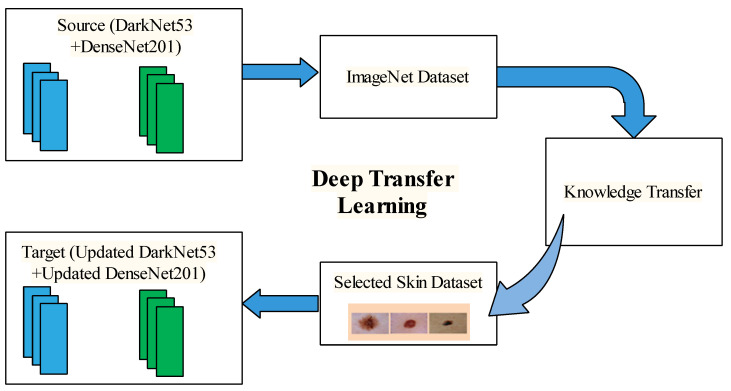
Transfer learning model for the learning of deep model for skin lesion classification.

**Figure 5 diagnostics-13-03063-f005:**
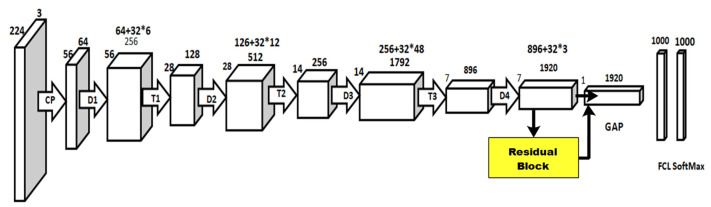
Original architecture of DensNet-201 CNN model.

**Figure 6 diagnostics-13-03063-f006:**
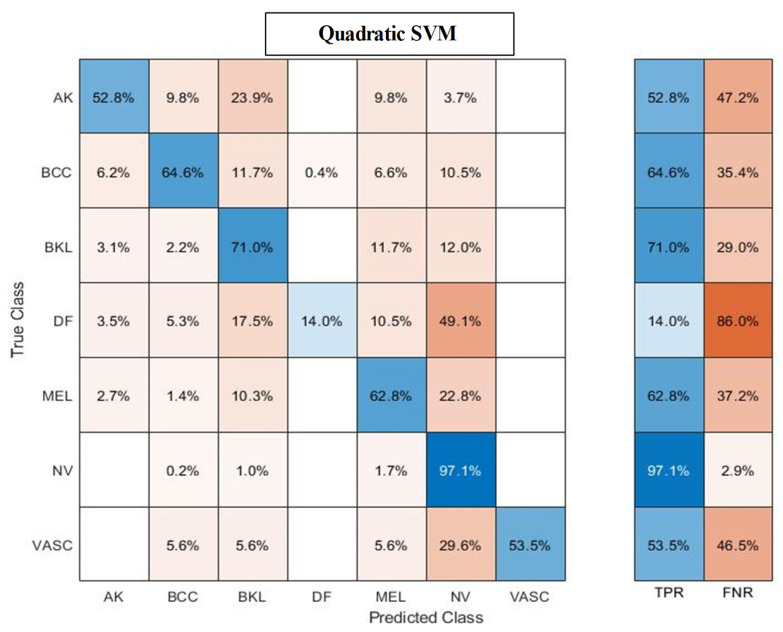
Confusion matrix of Quadratic SVM after employing the proposed feature selection technique on ISIC2018 dataset. * Actinic keratosis (AK), Melanoma (MEL), Melanocytic nevus (NV), Basal cell carcinoma (BCC), Benign keratosis (BKL), Dermatofibroma (DF), and Vascular (VASC), respectively.

**Figure 7 diagnostics-13-03063-f007:**
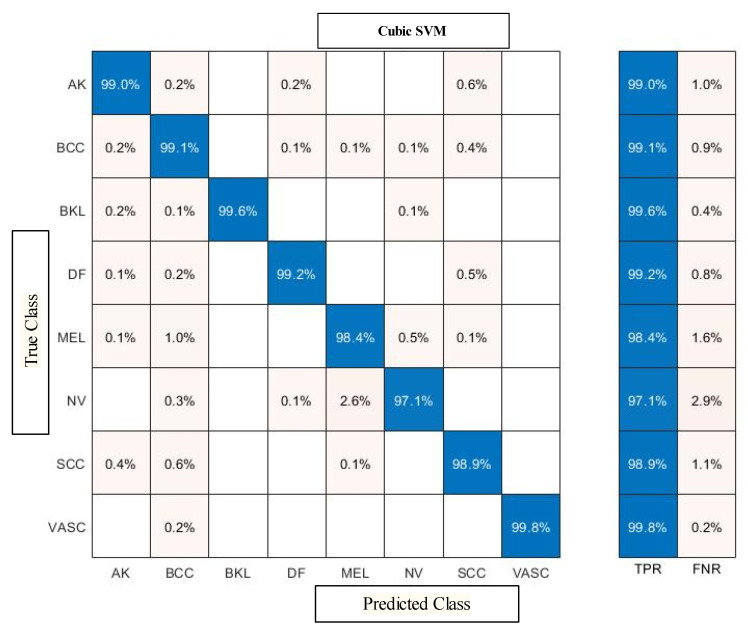
Confusion matrix of Cubic SVM after employing feature selection technique on ISIC2019 dataset.

**Figure 8 diagnostics-13-03063-f008:**
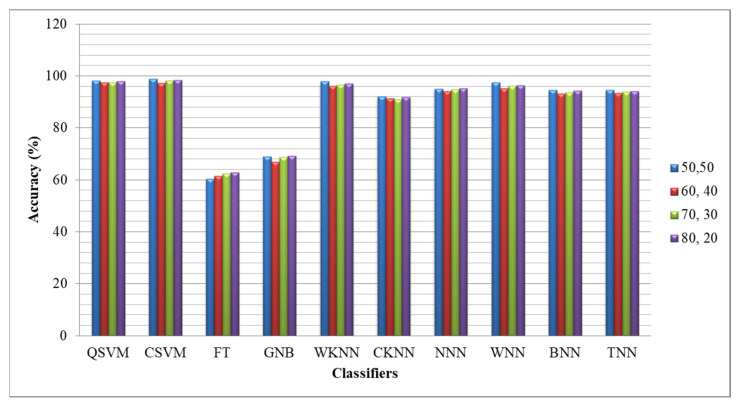
Comparison of ISIC2019 dataset accuracy after employing proposed feature selection using different training/testing ratios.

**Figure 9 diagnostics-13-03063-f009:**
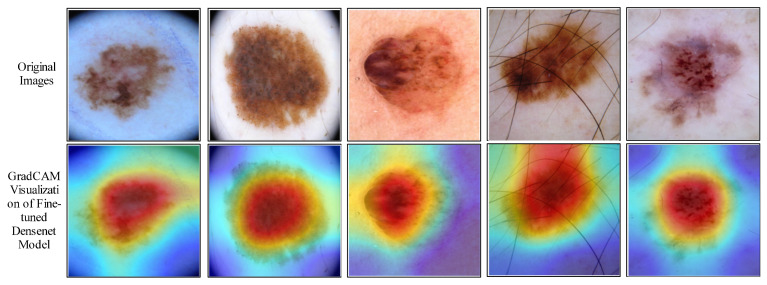
GradCAM based visualization of fine-tuned DenseNet-201model for cancer localization.

**Figure 10 diagnostics-13-03063-f010:**
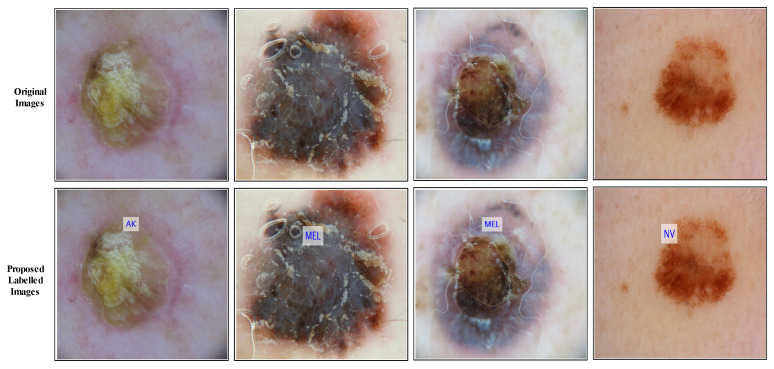
Proposed method prediction in terms of Labeled images.

**Table 1 diagnostics-13-03063-t001:** Summary of deep learning based classification technique.

Author	Year	Methods	Method Type	Dataset	Accuracy
Simon et al. [33]	2021	Interpretable deep learning framework	Detection + Classification	Private Collected	97.1%
Amin et al. [34]	2020	Alex net and VGG16 Neural Networks	Detection + Classification	Kaggle Skin Cancer	96.0%
Al-Masni et al. [35]	2020	ResNet-50 andDenseNet-201	Classification	ISIC 2016 ISIC 2017 and ISIC 2018.	88.0%
Pacheco et al. [36]	2020	SE Net with Adam Optimization	Detection + Classification	ISIC 2019	91.0%
Farooq et al. [37]	2019	Inception-V3 and Mobile Net Neural Networks	Classification	Kaggle Skin Cancer	86.0%
Liu et al. [38]	2019	Dense Net and Res Net use MFL module	Classification	ISIC 2017	87.0%
Pereira et al. [39]	2020	Linear SVM and Feedforward Neural Network (FNN)	Detection + Classification	Dermo fit Dataset	90.0%
El-Khitib et al. [41]	2020	Res Net-101 CNN Architecture	Detection + Classification	PH2 Dataset	90.0%
Almaraz et al. [42]	2020	Handcrafted features and Mobile Netv2 Architecture	Detection + Classification	HAM1000 Dataset	92.4%
Pacheco et al. [43]	2020	VGG-16, Mobile Net, Resnet-101 using clinical features	Classification	PAD-UFES-20	76.4%

**Table 2 diagnostics-13-03063-t002:** Proposed classification results by employing DarkNet-53 deep features on ISIC2018 dataset.

Sr.	Classifier (%)	Recall (%)	Precision (%)	F1 Score (%)	FNR (%)	Accuracy (%)	Time (Sec)
1	QSVM	49.7	72.9	59.1	27.1	79.0	109.98
2	CSVM	**49.2**	**72.7**	**58.6**	**27.3**	**79.3**	**114.2**
3	FT	25.9	31.6	28.4	68.4	66.6	12.44
4	GNB	49.4	33.3	39.7	66.7	55.3	24.2
5	WKNN	34.9	64.5	45.2	35.5	73.7	27.3
6	CKNN	72.2	46.7	56.7	53.3	72.2	467.3
7	NNN	72.9	48.0	57.8	52	72.9	411.4
8	WNN	72.5	46.2	56.4	53.8	72.5	371.6
9	BNN	76.3	55	63.9	45	76.3	26.7
10	TNN	71.4	41.7	52.6	58.3	71.4	345.16

Bold denotes the max values.

**Table 3 diagnostics-13-03063-t003:** Results of classification utilizing DarkNet-201 deep features using the ISIC2018 dataset.

Sr.	Classifier (%)	Recall (%)	Precision (%)	F1 Score (%)	FNR (%)	Accuracy (%)	Time (Sec)
1	QSVM	53.3	73.9	61.9	26.1	81.3	228.24
2	CSVM	**53.8**	**74.5**	**62.4**	**25.5**	**81.5**	**259.6**
3	FT	27.5	32.6	29.8	67.4	66.5	24.6
4	GNB	49.9	31.8	38.8	68.2	59	66
5	WKNN	35.9	67.5	46.8	32.5	74.7	54.172
6	CKNN	32.9	51.1	40.0	48.9	73.5	1103.2
7	NNN	49.6	50.3	49.9	49.7	75.1	604.1
8	WNN	56.9	56.9	56.9	43.1	79.3	653.42
9	BNN	46.5	46.5	46.5	53.5	74.4	52.0
10	TNN	42.7	42.3	42.4	57.7	72.8	253.2

Bold denotes the max values.

**Table 4 diagnostics-13-03063-t004:** Classification results of the proposed feature fusion technique using the ISIC2018 dataset.

Sr.	Classifier (%)	Recall (%)	Precision (%)	F1 Score (%)	FNR (%)	Accuracy (%)	Time (Sec)
1	QSVM	**61**	**80**	**69.2**	**20**	**86.2**	**448.7**
2	CSVM	59	81	68.2	19	86.1	552.49
3	FT	31	34	32.4	66	69.6	66.55
4	GNB	58	43	49.3	57	68.3	150.82
5	WKNN	39	48	43.0	52	77.1	96.49
6	CKNN	35	54	42.4	46	75.9	2537.2
7	NNN	58.4	60	59.1	40	81.8	545.7
8	WNN	66.4	71.7	68.9	28.3	85.5	81.6
9	BNN	57.4	57.9	57.6	42.1	81.4	1031.6
10	TNN	57.4	56	56.6	44	80	839.5

Bold denotes the max values.

**Table 5 diagnostics-13-03063-t005:** Classification results of the proposed feature selection technique on the ISIC2018 dataset.

Sr.	Classifier (%)	Recall (%)	Precision (%)	F1 Score (%)	FNR (%)	Accuracy (%)	Time (Sec)
1	QSVM	**60.8**	**78.1**	**68.3**	**21.9**	**85.4**	**277.9**
2	CSVM	**59.5**	**79.2**	**67.9**	**20.8**	**85.4**	**23.0**
3	FT	29.3	32.2	30.6	67.8	68.8	36.6
4	GNB	57.5	44.4	50.1	55.6	68.5	56.5
5	WKNN	37.1	70.2	48.5	29.8	77.0	53.0
6	CKNN	36.8	69.3	48.0	30.7	75.9	821.0
7	NNN	58.8	59.4	59.0	40.6	81.3	191.5
8	WNN	65.4	70.9	68.0	29.1	84.6	40.6
9	BNN	54.3	56.95	55.5	43.05	80.6	406.9
10	TNN	50.8	52.6	51.6	47.4	79.2	393.3

Bold denotes the max values.

**Table 6 diagnostics-13-03063-t006:** Classification results of DarkNet-53 deep features using the ISIC2019 dataset.

Sr.	Classifier	Recall (%)	Precision (%)	F1 Score (%)	FNR (%)	Accuracy (%)	Time (Sec)
1	QSVM	96.96	97.1	97.0	2.9	97.0	267.3
2	CSVM	**98**	**98.2**	**98.0**	**1.8**	**98.1**	**267.8**
3	FT	56.1	59.1	57.5	40.9	56.2	24.1
4	GNB	59.7	62.7	61.1	37.3	58.2	58.2
5	WKNN	97.7	98.1	97.8	1.9	98.0	112.63
6	CKNN	91.3	92.3	91.7	7.7	91.9	2471.6
7	NNN	95.7	95.8	95.7	4.2	95.9	636
8	WNN	95.7	96	95.8	4	95.9	648.4
9	BNN	97	97.2	97.0	2.8	97.2	571.1
10	TNN	95	95.4	95.1	4.6	95.4	667.9

Bold denotes the maximum value.

**Table 7 diagnostics-13-03063-t007:** Results of DensNet-201 deep features using ISIC2019 dataset.

Sr.	Classifier	Recall (%)	Precision (%)	F1 Score (%)	FNR (%)	Accuracy (%)	Time (Sec)
1	QSVM	98.3	98.4	98.3	1.6	98.3	1055.2
2	CSVM	**98.9**	**98.9**	**98.9**	**1.1**	**98.9**	**1177.6**
3	FT	62.3	65.4	63.8	34.6	62.1	48.3
4	GNB	63.1	65.3	64.1	34.7	61.5	102.8
5	WKNN	98.5	98.8	98.6	1.2	98.7	306.1
6	CKNN	94.3	93.2	93.7	6.8	94.0	7719.7
7	NNN	96.7	96.9	96.79	3.1	96.9	1054.5
8	WNN	98.8	96.9	97.8	3.1	96.9	1092.7
9	BNN	98	98.1	98.0	1.9	98.1	1167
10	TNN	96.3	96.4	96.3	3.6	96.9	1092.6

Bold denotes the maximum value.

**Table 8 diagnostics-13-03063-t008:** Results of the proposed fusion technique using the ISIC2019 dataset.

Sr.	Classifier	Recall (%)	Sensitive (%)	F1 Score (%)	FNR (%)	Accuracy (%)	Time (Sec)
1	QSVM	**98.3**	**98.3**	**98.3**	**1.7**	**98.4**	**1053.5**
2	CSVM	**99.02**	**99.1**	**99.0**	**0.9**	**99.1**	**1329.7**
3	FT	62.5	65.4	63.9	34.6	62.5	128.2
4	GNB	70.5	73	71.7	27	69.1	176.3
5	WKNN	97.6	86.06	91.4	13.94	98.0	513.1
6	CKNN	91.1	93.2	92.1	6.8	92.7	1253.3
7	NNN	95.9	95.9	95.9	4.1	96.0	266.5
8	WNN	97.8	97.9	97.8	2.1	98.0	181.26
9	BNN	95.6	95.7	95.6	4.3	95.7	414.8
10	TNN	95.3	95.3	95.3	4.7	95.3	610.83

Bold denotes the maximum value.

**Table 9 diagnostics-13-03063-t009:** Results of proposed feature selection on the ISIC2019 dataset.

Sr.	Classifier (%)	Recall (%)	Sensitive (%)	F1 Score (%)	FNR (%)	Accuracy (%)	Time (Sec)
1	QSVM	98.1	98.2	98.1	1.8	98.2	488.8
2	CSVM	**98.8**	**98.9**	**98.8**	**1.1**	**98.9**	**655.1**
3	FT	60.6	63.2	61.8	36.8	60.4	39.9
4	GNB	70.3	72.5	71.3	27.5	68.9	87.39
5	WKNN	97.7	98	97.8	2	97.9	180.1
6	CKNN	91.8	91.8	91.8	8.2	92.1	3829.5
7	NNN	95.1	95.1	95.1	4.9	95.1	105.9
8	WNN	97.3	94.7	95.9	5.3	97.5	129.4
9	BNN	94.7	97.4	96.0	2.6	94.6	342.28
10	TNN	94.5	94.5	94.5	5.5	94.5	128.6

Bold denotes the maximum value.

**Table 10 diagnostics-13-03063-t010:** Comparison with existing methods for the proposed technique.

Reference	Dataset	Accuracy
[49]	ISIC 2018	83%
[50]	ISIC 2019	97.1%
[51]	ISIC 2019	97.84%
**Proposed**	ISIC 2018 ISIC 2019	**85.4%** **98.9%**

Bold denotes the significant outcome.

## Data Availability

The datasets used in this work are publically available (https://challenge.isic-archive.com/data/#2019, accessed on 11 August 2023).

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
