# Peer review of "MSRNet: Multiclass Skin Lesion Recognition Using Additional Residual Block Based Fine-Tuned Deep Models Information Fusion and Best Feature Selection"

_diagnostics, 2023, doi:10.3390/diagnostics13193063_

Round 1

Reviewer 1 Report

Authors developed a new deep-learning based tool for skin cancer.  Although there are much of technical details, there are several key information missing, which makes the evaluation of authors’ claim virtually impossible.    

Major comments

1.       Lines 188-194: Information about the images are missing.  What are the sources?  How was the normalization done (choice of min/max intensity within image and size)?  How does this change if dealing with 8-, 16-, 32- or 64-bit images?      

2.       Lines 240-265: There are no references regarding DarkNet-53 and DensNet-201, and it is not clear what is the input/output of these models.

3.       Line 270: What is R?

4.       Lines: 361-366: How was the classification done?  How many classes?

5.       Lines 374-375: Is there a reason that authors do not consider validation data?

6.       Table 2: Please, elaborate on how the accuracy was calculated (based on ISIC2018 classification, DarkNet53 classification, or some other clinical information). 

7.       How will be the resources publicly available?

Minor comments

1.       The resolution of Fig 1 is low.

2.       Line 369: Please, state the total number of images in the analyses.

The manuscript appears to be well written.

Author Response

Dear Reviewer, thank you for your comments. Please consider the revised version for your review. thanks

Reviewer 2 Report

The topic explored by the authors is interesting and the overall work is well presented. I have only some minor remarks:

- please carefully check for typos, also in the figures: es. in Figure 1 "optiomization", "prediatior"

- to improve readability for general clinicians, some paragraphs in the introductory parts could be added to briefly and clearly explain what the algorithms and the databases used are

- please add a legend for the classes reported in the figures with the confusion matrices: what do AK, BCC and so on stay for? As a clinician I can guess myself, but for readability and completeness it is required that all abbreviations are explained in footnote 

- in the conclusive parts I would like to read some considerations on the clinical applicability of the findings

No special remarks on the English language, only check for typos.

Author Response

(The authors gave the same response as above.)

Reviewer 3 Report

In this study, the authors developed the MRSNet model for skin lesion diagnosis. They applied a contrast enhancement technique for image preprocessing, made modifications to DarkNet53 and DenseNet201 to enhance their performance, and so on. The authors also described their steps and models in detail. However, several questions remain unclear, and the manuscript requires further consideration after addressing the following concerns.

1.    Throughout the manuscript, there are inconsistencies in the notation of neural network names, such as “Darknet53”, “DarkNet53”, “DensNet201”, “DensNet201”, “DenseNet201”. It is essential to maintain consistent and accurate naming conventions without typographical errors.

2.    Definitions for abbreviations such as "BN" (Batch Normalization) in line 247 and "TL" (Transfer Learning) in line 252 should be provided to enhance reader comprehension.

3.    The authors split training and testing data into 50:50, deviating from typical split ratio 8:2. It would be beneficial to elucidate the specific reasons for choosing this unconventional split ratio.

4.    Line 434 mentions “DenseNet201 model is improved than the DarkNet model.” This improvement should be elucidated, whether it results from a deeper architecture, unique structural elements in DenseNet201, or other factors not present in DarkNet.

5.    Line 445 states mentioned “1% improvement in the accuracy is observed”. When authors have different training trials, what is the standard deviation for accuracy, is 1% improvement in the accuracy significantly?

Thank you for the opportunity to review this manuscript. Overall, I find their research was well-conducted and the manuscript is well-written. Overall, I think this research is valuable and important to the field. With some revisions and improvements, this manuscript has the potential to be a good publication.

1.    In line 198, “I(u,v) the original RGB is an image”, what does this mean?

2.    In line 452, “Figure 6 showing the confusion matrix of Cubic SVM for the feature selection results.” Please check the grammar.

Author Response

(The authors gave the same response as above.)

Round 2

Reviewer 1 Report

Manuscript has improved.